# In Vitro Evaluation of Kaempferol-Loaded Hydrogel as pH-Sensitive Drug Delivery Systems

**DOI:** 10.3390/polym14153205

**Published:** 2022-08-05

**Authors:** Qin Zhang, Xinying Yang, Yifang Wu, Chang Liu, Hongmei Xia, Xiaoman Cheng, Yongfeng Cheng, Ying Xia, Yu Wang

**Affiliations:** 1College of Pharmacy, Anhui University of Chinese Medicine, Hefei 230012, China; 2Clinical College of Anhui Medical University, Hefei 230601, China; 3School of Life Science, University of Science and Technology of China, Hefei 230027, China

**Keywords:** kaempferol, carbopol 934, antioxidation, different pH, drug release

## Abstract

The purpose of this study was to prepare and evaluate kaempferol-loaded carbopol polymer (acrylic acid) hydrogel, investigate its antioxidant activity in vitro, and compare the effects on drug release under different pH conditions. Drug release studies were conducted in three different pH media (pH 3.4, 5.4, and 7.4). The kaempferol-loaded hydrogel was prepared by using carbopol 934 as the hydrogel matrix. The morphology and viscosity of the preparation were tested to understand the fluidity of the hydrogel. The antioxidant activity of the preparation was studied by scavenging hydrogen peroxide and 2,2-diphenyl-1-picrilhidrazil (DPPH) radicals in vitro and inhibiting the production of malondialdehyde in mouse tissues. The results showed that kaempferol and its preparations had high antioxidant activity. In vitro release studies showed that the drug release at pH 3.4, 5.4, and 7.4 was 27.32 ± 3.49%, 70.89 ± 8.91%, and 87.9 ± 10.13%, respectively. Kaempferol-loaded carbopol hydrogel displayed greater swelling and drug release at higher pH values (pH 7.4).

## 1. Introduction

Kaempferol (KAE) is a representative natural flavonoid compound, which widely existing in a variety of edible plants, vegetables, and fruits [1], such as kaempferia galanga, ginger, leek, etc. KAE has also been found in several medicinal plants, including *Ginkgo biloba*, *Ziziphus mauritiana,* and *Sophora japonica* [2]. It can be extracted from *kaempferia galangal*, a medicinal and edible plant, and mainly exists in its roots and rhizome. *Kaempferia galanga* has obvious beneficial effects on human health and can be used for treating thoracic and diaphragmatic distention, epigastric and abdominal cold pain, and dietary consumption. Flavonoids had uniform structural characteristics, a skeleton consisting of 15 carbon atoms and two aromatic rings connected by an oxygen-containing heterocyclic ring [3]. The pyran ring was the most typical feature of every flavonoid subfamily. Flavonoids are similar in structures, but small structural differences can lead to great changes in their biological characteristics, and the position and number of hydroxyl groups determine their antioxidant activity intensity [4]. KAE had four hydroxyl groups, so it had strong antioxidant capacity. KAE as a drug has attracted wide attention because of its strong antioxidant activity, which is beneficial to human health and is a common antioxidant in our daily diet [5]. Studies have shown that KAE has a wide range of pharmacological activity, such as antioxidant [6], anticancer [7], anti-inflammatory [8], anti-platelet aggregation [9], anti-diabetes [10], anti-obesity [11], anti-allergic [12], and antibacterial activity [13], as well as the prevention of bone disease [14] and cardiovascular protection [15]. KAE has hydrophobic properties due to its diphenyl propane structure, which leads to poor water solubility and reduced bioavailability in vivo, thus limiting its clinical application [16,17].

The hydrogel as an excellent drug carrier can form a high chemically cross-linked network structure with hydrophilic three-dimensional systems [18], and it has unique properties [19], such as good biocompatibility, fluidity, bioadhesion, sustained release, reducing drug toxicity, improving drug stability, and so on. Carbopol 934 (CBP) is a polymer crosslinked by polyacrylic acid showing a highly porous morphology, which has sensitive responses to different pH values. As a drug carrier, CBP hydrogel has a certain sustained-release effect on the drug, which can improve the concentration and bioavailability of drugs in blood. It has been widely used in the drug and medical fields due to its unique characteristics, such as easy deformation, transparent porosity, a certain viscosity, and adhesion [20]. The CBP hydrogel is pH sensitive and very suitable for acidic tumor microenvironment, which might be a good drug delivery system for selective release of antitumor drugs without damaging normal tissues. The hydrogel can also be used as a drug carrier to treat inflammation, such as drug delivery of nepafenac after cataract surgery, to treat postoperative inflammation [21] and to promote targeted delivery of budesonide to treat chronic inflammatory bowel disease [22]. Since CBP hydrogel was a copolymer of acrylic crosslinked polymer, its molecule contained a large number of carboxyl groups. Under different pH conditions, the ionizable groups existing in the molecule are different. With the different pH of the medium, the drug will be intelligently released. Imran Kazmi et al. [23] studied the treatment of skin cancer with luteolin-loaded CBP hydrogel. Maimoona Qindeel et al. [24] studied CBP hydrogel loaded with methotrexate for the treatment of rheumatoid arthritis. Therefore, CBP hydrogel has the potential to be used as a drug release carrier for tumor and inflammatory diseases.

In the present study, the pH-sensitive CBP hydrogel was selected as the drug carrier of KAE, and its antioxidant activity in vitro was studied, and the drug release under different pH conditions was compared. The antioxidant activity of kaempferol-loaded hydrogel was examined in vitro by the free radical scavenging activities on hydrogen peroxide and DPPH, and the drug was released in vitro into media with different pH values. The release kinetics of kaempferol-loaded hydrogel was investigated by simulating various kinetic models.

## 2. Materials and Methods

### 2.1. Materials

Kaempferol and Carbopol 934 was purchased from Macklin Biochemical Company (Shanghai, China) with a purity of 97%, absolute ethanol was acquired from RichJoint Chemical Reagents Company (Shanghai, China), 1,1-diphenyl-2-picrylhydrazyl (DPPH) was purchased from Yuanye Biology Science and Technology Company (Shanghai, China) with a purity of 98%, FeSO_4_·7H_2_O was purchased from Sinopharm Chemical Reagent Company (Shanghai, China), 2-thiobarbituric acid (TBA) was purchased from Yuanye Biology Science and Technology Company (Shanghai, China) with a purity of 98%, trichloroacetic acid (TCA) was acquired from Damao Chemical Reagent Factory (Tianjin, China). All chemical reagents were of analytical grade.

### 2.2. Animals

The healthy Kunming female mouse (20 ± 2 g) was purchased from the Animal Experimental Center of Anhui University of Chinese Medicine (Hefei, China). All animal experiments were carried out according to the guidelines approved by the Ethics Committee of Anhui University of Chinese Medicine (Hefei, China).

### 2.3. Methods

#### 2.3.1. Standard Curve of KAE

We accurately weighed 0.01 g KAE in a 10 mL test tube, dissolved it with the mixed solution of absolute ethanol and phosphate buffered saline solution according to the volume ratio is 1:4 (PBS, pH 7.4), and fixed the volume. Using it as the mother liquor, it was prepared into five concentrations of 0.0125, 0.025, 0.05, 0.1, and 0.2 mg/mL. Then, the absorbance of the solution at 360 nm was determined via the UV-spectrophotometer (1600 UV-Vis, Shanghai Mepeda instrument Co., Ltd., Shanghai, China), recorded, and established as a standard curve. A concentration of KAE was selected for precision and stability determination.

#### 2.3.2. Preparation of KAE-Loaded Hydrogel

The preparation process is shown in Figure 1. First, we accurately weighed 0.10 g of CBP, sprinkled it into 10 mL 2.5 mg/mL KAE solution, placed it on a magnetic stirrer, stirred at 37 °C, and left it to fully swell overnight to obtain KAE-loaded hydrogel (KAE-GEL). The KAE-LP prepared by PBS was used in antioxidant experiment. In the in vitro diffusion experiment, because of the poor permeability of KAE, 0.6% CBP was used as hydrogel matrix, absolute ethanol and glycerol were used as solvents (volume ratio 1:9), and 10% glycerol in PBS was used as release medium. Blank hydrogel (BLK-GEL) was prepared in the same way without KAE.

#### 2.3.3. Measurement of Viscosity of KAE-GEL

The viscosity of KAE-GEL was measured by Ubbelohde capillary viscometer. At room temperature, 1 mL of BLK-GEL and KAE-GEL was placed in a test tube, and then 9 mL of PBS was added and mixed. After mixing, we measured with capillary viscometer, recorded the time, and performed three parallel operations. The viscosity of hydrogel was calculated by using Equation (1):(1)$$\eta_{2}=\frac{\rho_{2}t_{2}}{\rho_{1}t_{1}}\times \eta_{1}\;$$
where “*η*” represents the viscosity (mpa·s), “*ρ*” represents the density (g/mL), and “*t*” represents the time (s); “1” represents the water and “2” represents the sample.

#### 2.3.4. H_2_O_2_ Free Radicals Scavenging Assay

When determining the scavenging activity of KAE-GEL on H_2_O_2_, three groups of experiments should be carried out simultaneously as follows. The control group was a mixture of 0.6 mL PBS and 1.8 mL H_2_O_2_ solution (40 mmol/mL), the sample group was a mixture of 0.6 mL samples and 1.8 mL H_2_O_2_ solution, and the sample control group was a mixture of 0.6 mL sample and 1.8 mL PBS. After the three groups of samples were mixed evenly, they were incubated at 37 °C in the dark for 10 min. Thereafter, the absorbance of incubation solution was measured at 230 nm. The H_2_O_2_ radical scavenging capacity (*RSC*) was calculated by Equation (2):(2)$$RSC\%=\left[1-\frac{A_{s}-A_{c}}{A_{o}}\right]\times 100\%\;$$
where “*A*_o_” represents the absorbance of blank control group, “*A*_s_” represents the absorbance of sample group, and “*A*_c_” represents the absorbance of sample control group.

#### 2.3.5. DPPH Free Radicals Scavenging Assay

When determining the scavenging activity of KAE-GEL on DPPH free radicals, three groups of experiments should be carried out simultaneously. The blank control group was 2 mL of absolute ethanol mixed with 1 mL of DPPH absolute ethanol solution (0.08 mg/mL), the sample group was 2 mL of sample mixed with 1 mL of DPPH absolute ethanol solution, and the sample control group was 2 mL of samples mixed with 1mL of absolute ethanol. After the three groups of samples were mixed evenly, the samples were incubated for 30 min in the dark at 37 °C. Thereafter, the absorbance of the incubation solution was measured at 517 nm. The DPPH radical scavenging rate was calculated as shown in Equation (2).

#### 2.3.6. Inhibitory Effect of KAE-GEL on Lipid Peroxidation of Mouse Tissues Ex Vivo

Preparation of tissues homogenate was completed as follows: the mouse was fasted for 12 h and killed after anesthesia. We quickly took out the liver, kidney, lung and brain tissue; rinsed them repeatedly with cold normal saline; washed off blood stains, blotted them with filter paper; weighed them; added nine times of cold normal saline; crushed them with a homogenizer; centrifuged at 4000 rpm for 15 min; and took the supernatant to make 10% tissues homogenate.

We took 1 mL of 10% fresh tissue homogenate, added 0.1 mL to the sample group, added the same volume of normal saline to the negative and positive control groups, and let it stand for five minutes. Then, 0.1 mL FeSO_4_ (0.278 g in 100 mL of distilled water) was put into the sample and positive control group for induction, while normal saline was added to the negative control group as blank control, which was heated in a water bath at 37 °C for 1.5 h. After taking it out, 2 mL TCA (2.8 g in 50 mL of distilled water) was added, left to stand for 5 min, and then we added 1mL TBA (0.375 g in 100 mL of 50 mmol/L NaOH); the sample was mixed well, put in a water bath at 95 °C for 40 min, cooled with running water, centrifuged at 4000 rpm for 8 min, and the supernatant was sucked. The absorbance was measured at 532 nm and was zeroed with normal saline. The inhibition rate (*E*) of KAE and KAE-GEL on the production of MDA in liver, kidney, lung, and brain tissue was calculated by Equation (3):(3)$$E\%\;inhibition=\left[\frac{A_{m}-A_{s}}{A_{m}-A_{b}}\right]\times 100\%\;$$
where “*A**_m_*” represents the absorbance of model group, “*A**_b_*” represents the absorbance of blank group, and “*A_s_*” represents the absorbance of sample group.

#### 2.3.7. Diffusion across the Dialysis Membrane In Vitro with Different pH

The Franz diffusion cells were used for the experiment (Figure 2). Because KAE was not easily soluble in PBS, but easily soluble in glycerol, a certain proportion of glycerol was selected as the release medium for in vitro diffusion, and on this basis, the corresponding pH was adjusted with NaOH and phosphoric acid. A diffusion bottle with a volume of 16 mL was used, and a small stirring magnet was placed in each diffusion bottle and filled with 37 °C glycerol solution. The dialysis membrane (MW: 8000–14,000) was soaked in boiling water, then cut with scissors, and the dialysis membrane was placed between the upper and lower cells of diffusion bottles, respectively, so that the dialysis membrane was in full contact with the glycerol solution (pH 3.4, pH 5.4, and pH 7.4, respectively) in lower cells without bubbles. The diffusion bottles were placed in culture dish and on a magnetic stirrer to continuously stir for thorough mixing of the solution of the lower cells. Then, 1 mL of sample solution was added into the upper cells of each diffusion bottle. Under the constant temperature of 37 ± 1 °C, continuous stirring was performed, and diffusion was performed for 84 h. Next, 2 mL samples were collected at 5 min, 10 min, 20 min, 30 min, 1 h, 2 h, 3 h, 4 h, 5 h, 6 h, 7 h, 8 h, 9 h, 10 h, 11 h, 12 h, 24 h, 36h, 48 h, 60 h, 72 h, and 84 h from lower cells, and we added the same volume of release medium to restore the original volume. Thereafter, their absorbance was measured at 360 nm and recorded. The same diffusion step was repeated three times. The cumulative permeation rate *Q_n_* (%) was calculated by Equation (4):(4)$$Q_{n}\left(\%\right)=\frac{\left(C_{n}\times V_{n}+\sum_{i=1}^{n-1}C_{i}\times V_{i}\right)}{C_{KAE/KAE-GEL}V_{KAE/KAE-GEL}}\times 100\%\;$$
where “*C**_i_*” and “*C**_n_*” represent the drug concentration in dissolution medium at each sampling and the last sampling point, respectively. “*V**_n_*” and “*V**_i_*” represent the volumes of the dissolution medium and the sample, respectively. “*C_KAE/KAE-GEL_V_KAE/KAE-GEL_*” is the total quantity of KAE, KAE-GEL.

#### 2.3.8. Drug Release Kinetic Study

Various mathematical models (zero order, first order, Higuchi, and Hixson-Crowell) [25,26] were used to determine the drug release kinetics and mechanism of KAE-GEL. The zero-order model was consistent with the release kinetics of an ideal drug release system, with a constant drug level during release, and was used to accumulate the percent drug released versus time. The first-order model was a semi-empirical kinetic model commonly used to simulate drug release from a porous structure and represents the cumulative percent of drug release versus the square root of time. The Higuchi model was used to study the release of water-soluble and low water-soluble protein in the matrix system and expressed the cumulative percent of drug release as a function of the square root of time. The Hixson-Crowell cube root model was used to express the relationship between the remaining percentage of drugs and the cube root of time. These kinetic models were used to analyze the cumulative release of KAE-GEL, and the correlation coefficient (R^2^) was the best simulation result. The formulas of various dynamic models are as follows:

Zero-order equation: $$M_{t}/M_{\infty }=kt$$

First-order equation: $$M_{t}/M_{\infty }=1-e^{k_{0}t}$$

Higuchi equation:$$M_{t}/M_{\infty }=k_{1}t^{1/2}$$

Hixson-Crowell equation: $$W_{0}{}^{1/3}-W_{t}{}^{1/3}=k_{2}t$$
where “*M_t_*/*M_∞_*” represents the cumulative percent released at time *t*, “*W*_0_” represents the initial amount of drug in the hydrogel, and “*W_t_*” represents the remaining amount of drug in the hydrogel at time t. “*k, k*_0_*,* and *k*_1_” are the zero-order, first-order, and Higuchi release constants, respectively, and “*k*_2_” is a constant that combines surface volume relation.

## 3. Results and Discussion

### 3.1. Standard Curve of KAE

The standard curve was plotted with the mass concentration (C) of KAE as the horizon coordinate and the absorbance (A) as the vertical coordinate, and the linear regression equation of A = 0.1434 × C + 0.0788 (R^2^ = 0.9992) was obtained. It showed that there has a good linear relationship between KAE concentration and absorbance in the range of 0.0125−0.2 mg/mL. The results showed that KAE precision and stability was good, RSD was less than 1%, and the solution was stable.

### 3.2. Structural Analysis of KAE and KAE-GEL

KAE was a natural phenolic phytochemical and a secondary metabolite in plants. It formed antioxidant structures in plants through a series of biosynthesis (Figure 3A). Firstly, three molecules of malonyl-CoA and one molecule of 4-coumaroyl-CoA were catalyzed by chalcone synthase to produce naringenin chalcone. Secondly, under the catalysis of chalcone isomerase, naringenin chalcone was transformed into naringenin flavonoids by closing the C3 ring. Then, under the catalysis of flavone 3-dioxygenase, a hydroxyl group was added to the C3 ring of naringenin to form dihydrokaempferol. Finally, a double bond was introduced into the C2–C3 region of dihydrokaempferol by flavanol synthase [27,28]. As three phenolic hydroxyl groups, one enol type and one carbonyl group were synthesized in plants, the antioxidant effect was exerted. The hydroxyl-containing structure at C3, C5, C7, and C4’, a carbonyl group at C4 and a double bond at C2–C3 are important structures, which make KAE own the characteristics of antioxidant activity [29,30]. As the oxygen atom of phenolic hydroxyl adopts sp2 hybridization, a pair of lone electrons are provided to form a delocalized bond with the six carbon atoms of the benzene ring, a large π bond strengthens the acidity, and the electron-donating effect of the hydroxyl further strengthens the polarity of the O-H bond so that hydrogen in phenol can be ionized out and is easily oxidized. There are enol forms on C2–C3, which are extremely unstable, dehydrogenated atoms. Therefore, the antioxidant effect of KAE is mainly manifested as the reaction of phenolic hydroxyl and enol-form hydroxyl with free radicals, removing hydrogen atoms and combining with free radicals to form a stable structure, thus terminating the chain reaction of free radicals. CBP was a hydrophilic polyacrylic acid polymer, whose carboxyl group was highly ionized after neutralization, and forms a hydrogel due to the electrostatic repulsion between charged polymer chains [31]. In the dry powder state, the carbon chains were tightly aggregated. After being dispersed into water, the molecules were gradually combined with water, so that the tightly aggregated carbon chains began to loosen, and the viscosity of the corresponding dispersion system increased. When KAE was added into CBP hydrogel, strong hydrogen bonds are formed. As shown in Figure 3B, KAE as a hydroxyl donor and CBP hydrogel as a carboxyl donor can combine with one to two or more hydroxyl groups to form hydrogen bonds. Due to intermolecular interactions (mainly hydrogen bonding), KAE was uniformly dispersed in the stereospecific structure of the CBP hydrogel.

### 3.3. Viscosity of KAE-GEL

Using 1% CBP as the matrix of hydrogel, the viscosity of KAE-GEL was moderate, the fluidity was good, and the whole system was relatively uniform and stable; it was a colorless, tasteless, and transparent hydrogel. The characteristics of these preparations are shown in Table 1.

### 3.4. Scavenging Ability of KAE-GEL on H_2_O_2_ Assay

As shown in Figure 4A, KAE had obvious scavenging effect on H_2_O_2_, which was not very stable and easy to decompose. The oxygen bond in the molecular structure was easy to break and form oxygen hydrogen free radicals. Therefore, hydrogen peroxide was a free radical generator. In the 0.1–0.5 mg/mL range of KAE concentration, the scavenging effect of KAE on hydroxyl radicals increased with the increase of concentration, showing a dose–effect relationship. The experimental results showed that the IC_50_ value of KAE on hydroxyl radical scavenging rate was about 0.5 mg/mL. Since KAE was encapsulated in hydrogel, the H_2_O_2_ scavenging ability of KAE-GEL was weaker than that of KAE, and the scavenging rate on hydroxyl radicals was 29.12 ± 3.17% for KAE and 18.02 ± 1.53% for KAE-GEL (Figure 4B). The scavenging capacity of KAE on hydroxyl free radicals was about 1.5 times that of KAE-GEL. These results showed that KAE in KAE-GEL was encapsulated by the hydrogel and did not fully exert its scavenging ability on free radicals, but it also had some scavenging ability and would continuously release the drug to exert its effect.

### 3.5. Scavenging Ability of KAE-GEL on DPPH Assay

Among the methods for determining antioxidant activity, the DPPH method is one of the effects and most widely used. It is a nitrogen-centered free radical [32]. The mechanism (Figure 4C) involves the reduction of DPPH free radical by hydrogen atom transfer, which causes the color to change from deep purple to light yellow [33,34]. After being reduced by antioxidants, the DPPH radical becomes stable molecules, which can be measured at 517 nm [35] to determine the activity of antioxidants. KAE had obvious DPPH free radical scavenging ability in the experiment, as shown in Figure 4D. Because KAE has the ability to scavenge free radicals, Keti Zeka et al. [36] selected polyvinyl pyrrolidone hydrogel as the drug release system for wound treatment or beautification. In the concentration range of 0.0004–0.25 mg/mL, its scavenging effect increases with the increase of dose, and the color of DPPH gradually turns pale yellow with the increase of concentration. When the concentration of KAE is above 0.01 mg/mL, its free radical scavenging ability on DPPH tends to be saturated. Sui-Ping Deng et al. studied that kaempferol has a strong DPPH scavenging ability [37]. As shown in Figure 4E, in our study, the scavenging ability of 0.01 mg/mL of KAE and KAE-GEL on DPPH was 91.52 ± 8.88% and 41.57 ± 2.19%, and the scavenging ability of KAE-GEL on DPPH radicals was lower than that of KAE-GEL. This may be because KAE was wrapped into the CBP hydrogel and was not complete released. In hydrogel preparation, because of the three-dimensional structure and water solubility of hydrogel, its scavenging ability was lower than that of KAE.

### 3.6. Effect of KAE-GEL on Peroxidation of Isolated Mouse Tissues by Malondialdehyde Colorimetry of Tissues Homogenate

In the process of lipid peroxidation (Figure 5A), reactive oxygen species (ROS) with biological macromolecules, such as phospholipids, enzymes, and membrane receptor-related polyunsaturated fatty acids and nucleic acids, to form lipid peroxidation products, such as malondialdehyde (MDA) and 4-hydroxynonenoic acid [38]. MDA was one of the common markers of lipid peroxidation products and is also a widely studied lipid peroxidation product. It is a CH-acidic dicarbonyl compound, a protic acid in aqueous solution with a pKa of 4.46, and has the same acidic properties as aliphatic carboxylic acids [39]. Due to its carbonyl function, MDA can undergo chemical reactions, which are prone to polymerization and some reactions with the nucleophilic centers of various biomolecules. The chemical analysis of MDA begins with its determination as a component of the thiobarbituric acid reactive species for the evaluation of lipid peroxidation. Under acidic and high temperature conditions (95 °C) for an extended reaction time (40 min), one molecule of MDA reacted with two molecules of TBA to form a red derivative, commonly abbreviated as MDA-TBA, with strong visible light absorption at 532 nm (Figure 5B).

In the lipid peroxidation experiments of isolated mouse tissues, we added Fe^2+^ as induction, and the colors of the negative control and positive control groups in Figure 6A were significantly different; additionally, the color of the negative control group was darker than that of the positive control group, indicating that Fe^2+^ induced more MDA production in the tissues. More MDA was produced in isolated liver, kidney, lung, and brain tissues of mice. The MDA content in each tissue could be observed from the color of the homogenate: brain tissue > kidney tissue >liver tissue > lung tissue. With the addition of 0.1 mg/mL of KAE and KAE-GEL, the four tissues exhibited different inhibition results. The inhibitory rate of KAE on MDA production in brain, kidney, liver, and lung tissue was 43.7 ± 4.23%, 33.54 ± 3.47%, 29.2 ± 3.12%, and 18.1 ± 1.8%, respectively. It could be seen that KAE had the best inhibitory effect on MDA in brain tissue and that KAE-GEL had a stronger inhibition effect than BLK-GEL from Figure 6B. The inhibition rates of MDA production by BLK-GEL in brain, kidney, liver, and lung tissues were 26.6 ± 2.7%, 22.32 ± 2.37%, 15.88 ± 1.79%, and 6.49 ± 0.69%, respectively. This indicates that CBP hydrogel also had an inhibitory effect on lipid peroxidation. KAE-GEL definitely had a stronger ability to inhibit lipid peroxidation than BLK-GEL due to the encapsulation of KAE. The inhibitory rate of KAE-GEL on MDA production in brain tissue was 57.6 ± 5.96%, that in kidney tissue was 53.1 ± 5.55%, that in liver tissue was 35.8 ± 3.98%, and that in lung tissue was 19.2 ± 2.08%. The results showed that the formulation inhibited lipid peroxidation more than KAE and BLK-GEL, probably because of the synergistic effect of KAE and CBP hydrogel. It has been reported that KAE can scavenge different types of free radicals, promote the activity of antioxidant enzymes, and inhibit the production of ROS and lipid peroxidation [40]. The research by Saw et al. showed that the addition of KAE to HepG2-C8 cells treated with H_2_O_2_ could reduce the formation of ROS [41]. The results of this experiment also confirmed that KAE and KAE-GEL can reduce oxidative damage and MDA production in various tissues by reducing the production of ROS.

### 3.7. Results of KAE-GEL Diffusion In Vitro with Different pH

The dialysis membrane method was used to study the drug release in vitro. The principle was that the dialysis membrane can intercept molecules of a certain size. During dialysis, biological macromolecules in the sample solution are trapped on the dialysis membrane, while small molecular drugs continuously diffuse across the dialysis membrane and enter the lower buffer solution. In the diffusion process, the drug transports from the high concentration side to the low concentration side according to the concentration gradient difference. With the increase of diffusion time, KAE in the upper cells gradually diffused into the glycerol release medium in the receiving cells.

The Figure 7A showed the viscosity of KAE-GEL at different pH values. Significant differences in the viscosity of CBP hydrogel at different pH values were noted. The viscosity of CBP hydrogel increased with pH and reached its highest when it was neutralized to pH 5.5 or so. This was associated with an increase in ionization due to an increase in pH, which led to an increase in electrostatic repulsion between adjacent carboxyl groups and the subsequent expanded polymer network. A release study was carried out for KAE-GEL at pH 3.4, 5.4, and pH 7.4, respectively. As shown in Figure 7B, at pH 7.4, KAE-GEL provided greater drug release than pH 3.5 and 5.4. With the change of pH, CBP hydrogel showed different swelling behaviors, and the maximum swelling appeared at pH 7.4. Due to the pH-dependent swelling of CBP hydrogel, the pH-dependent drug release of KAE-GEL was observed. The CBP hydrogel was a high molecular polymer crosslinked by acrylic acid and allyl sucrose. Carboxyl-containing acrylic substances can be protonated at lower pH values and negatively charged at higher pH values. They are often used as pH-sensitive monomers. When that external pH changes, these groups will ionize and cause formation or rupture of hydrogen bond bonds between the molecular chains in the hydrogel network [42]. It causes sol-gel changes and exhibits pH-sensitive properties. Slight changes of external pH value may have a significant impact on the phase transition of the hydrogel. When pH was in the range of 3.4–5.4, the carboxyl groups in CBP hydrogel was not or partially dissociated, but protonated and conjugated with other counterions. Hydrogen bond interaction occurred between carboxyl groups and a small amount of hydration occurred. The whole molecular chain was in a contracted state, showing low swelling and high viscosity of the hydrogel at the macro level. At this time, KAE was released less from the CBP hydrogel; the drug release was 27.32 ± 3.49% at pH 3.4 and at pH 5.4 it was 70.89 ± 8.91%. When the pH was between 5.4 and 7.4, the carboxyl groups of CBP hydrogels were deprotonated, the free carboxyl groups were increased, and the hydrogen bond interactions were decreased. These free carboxyl groups generated electrostatic repulsion to each other and undergo substantial hydration. The hydrogel expanded from the initial contraction state, and its molecular volume increased rapidly, showing high swelling and low viscosity. At this time, the release of KAE from CBP hydrogels was very high, and the drug release was 87.9 ± 10.13% at pH 7.4.

Muhammad Suhail et al. [43] studied the drug release of carbopol 934/sodium polystyrene sulfonate-copoly hydrogel in two different media. The hydrogel showed low swelling and very low drug release at pH 1.2, while the gel showed maximum swelling and high drug release at pH 7.4. The results are consistent with our experimental results, which further support our study. At pH about 7.4, CBP hydrogel was highly swollen, and the drug was released rapidly. The pH of the microenvironment in tumor and inflammatory diseases was between 5 and 7, and the findings suggest that KAE-GEL releases more drugs at this pH. Cuihua Liu et al. investigated that KAE could attenuate skin damage in murine psoriasis by reducing gene expression of pro-inflammatory cytokines [44]. Dongmei Pan et al. [45] studied that KAE inhibits migration and invasion of fibroblast-like synovial cells in rheumatoid arthritis by blocking MAPK pathway activation. KAE has chemopreventive and chemotherapeutic activity against a variety of tumors, and a recent study reported that KAE is a nuclear receptor 4A1 (NR4A1, Nur77) ligand that inhibits rhabdomyosarcoma cells and tumor growth [46]. Tawoo Kim et al. found that KAE induced autophagy death of gastric cancer cells through IRE1-JNK-CHOP signaling pathway [47]. Under the pH of the microenvironment of these diseases, KAE-GEL will also release more drugs, so as to fully exert the pharmacological effects of KAE and improve its bioavailability. Therefore, KAE-GEL can be used as a new type of hydrogel to treat some slightly acidic diseases, such as inflammation and tumors. The pH sensitivity of CBP hydrogel maybe increases the therapeutic effect of KAE in some pathological tissues.

### 3.8. Kinetics of KAE Release from CBP Hydrogel

In order to determine the release mechanism of preparations in different pH solvents, four models were studied, namely zero-order, first-order, Higuchi, and Hixson-Crowel models. The results of correlation coefficient R^2^ of simulation curves of each model were shown in Table 2. The first-order and Higuchi models of drug release had a higher R^2^, suggesting that the drug release mechanism of KAE-GEL appears to be more amenable to first-order kinetic studies.

## 4. Conclusions

KAE has attracted many people’s attention because of its extensive pharmacological activities based on its structure. It has strong antioxidant activity, and it can be ingested from vegetables, fruits, and drinks every day to prevent diseases and strengthen our health because antioxidants help protect the immune system, effectively fight against diseases and inhibiting free radicals and anti-aging. In order to improve the poor water solubility of KAE, CBP hydrogel as a new diverse form was adopted in this experiment. The CBP was added into KAE solution as hydrogel matrix to prepare KAE-GEL. The antioxidant activity of KAE-GEL was determined by the experiments of scavenging H_2_O_2_ and DPPH free radical and inhibiting the production of MDA in mouse different tissues ex vivo, and the results showed that it had strong antioxidant activity. The controlled release effect of KAE-GEL under different pH conditions was determined by diffusion experiment. The results showed that KAE-GEL had a higher drug release rate at pH 7.4, but a lower rate at pH 3.4. Therefore, KAE-GEL can be used for the treatment of some inflammatory and tumor conditions due to its pH sensitivity, to improve the bioavailability of KAE, and give full play to its pharmacological effects in clinical application. In addition, more studies should be carried out to evaluate the anticancer and anti-inflammatory activities of KAE-GEL.

## Figures and Tables

**Figure 1 polymers-14-03205-f001:**
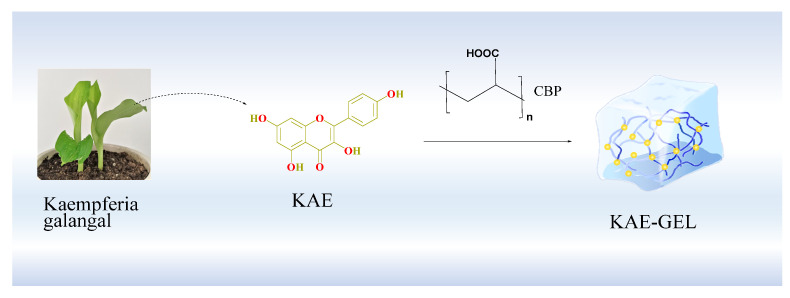
Preparation process of KAE-GEL.

**Figure 2 polymers-14-03205-f002:**
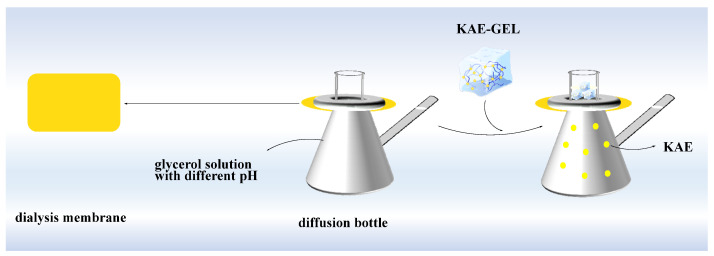
In vitro diffusion study of KAE-GEL through dialysis membrane (MW:8000–14,000) at 37 °C.

**Figure 3 polymers-14-03205-f003:**
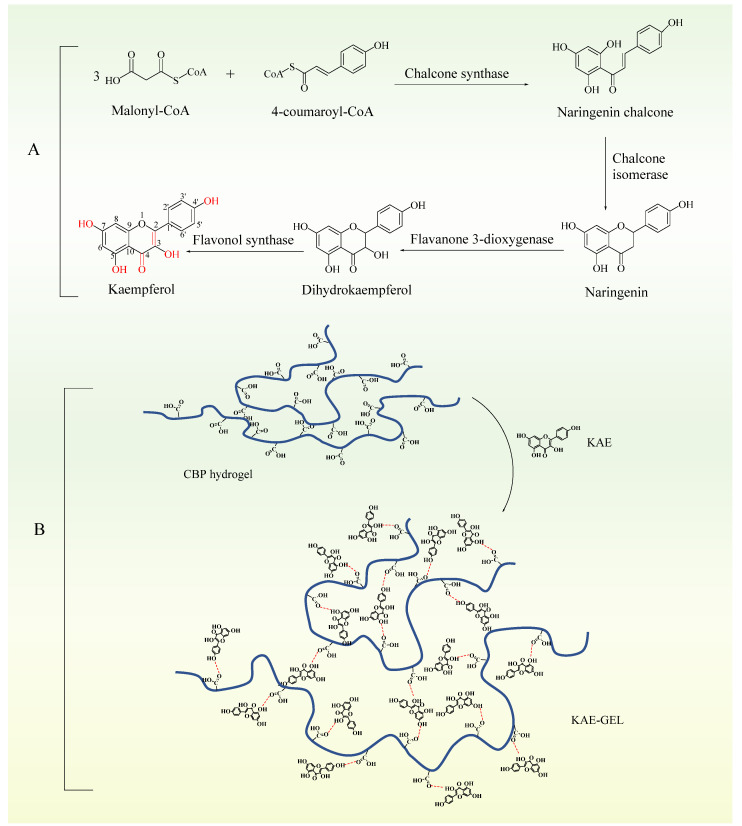
(**A**) Chemical and structural analysis of KAE. (**B**) Formation of KAE-GEL.

**Figure 4 polymers-14-03205-f004:**
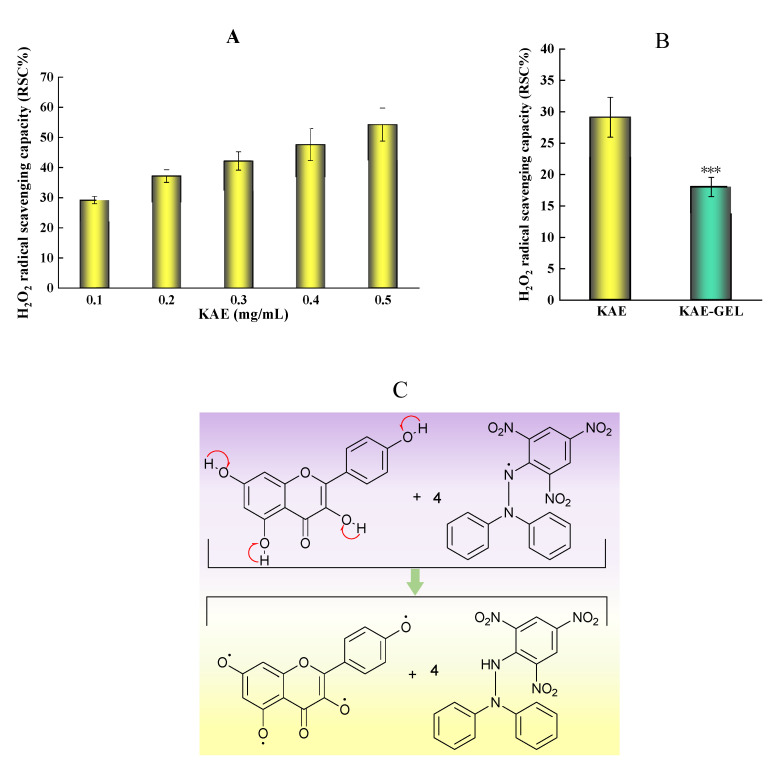
Free radical scavenging rate of KAE and KAE-GEL on H_2_O_2_ and DPPH. (**A**) Antioxidant activities of different concentrations KAE on H_2_O_2_. (**B**) Antioxidant activities of 0.1 mg/mL KAE and KAE-GEL on H_2_O_2_. (**C**) DPPH radical scavenging mechanism. (**D**) Different concentrations KAE express the scavenging ability on DPPH free radical. (**E**) 0.1 mg/mL KAE and KAE-GEL express the scavenging ability on DPPH free radical. This difference is statistically significant, *p* *** < 0.001.

**Figure 5 polymers-14-03205-f005:**
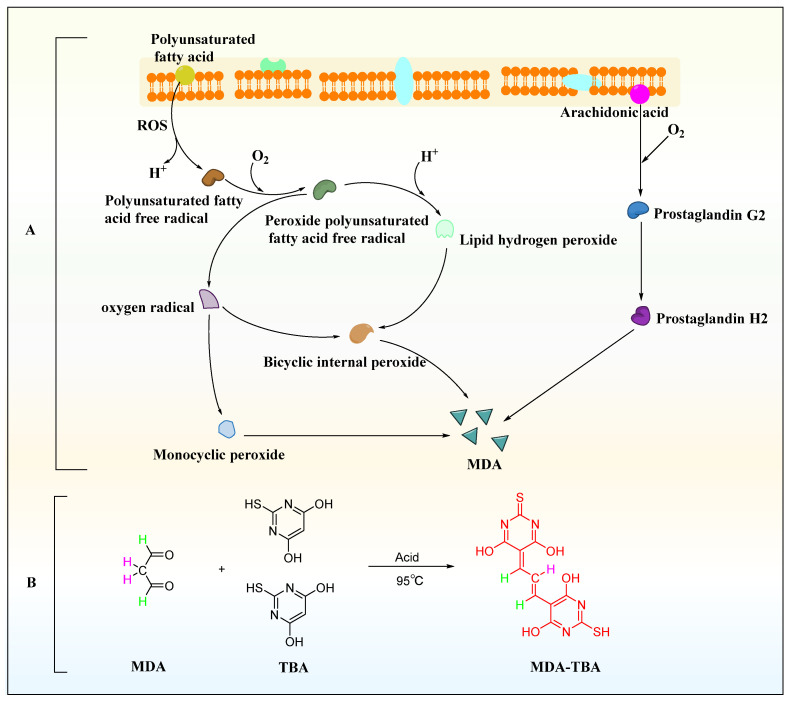
MDA generation and detection method. (**A**) MDA production process. (**B**) Thiobarbituric acid assay.

**Figure 6 polymers-14-03205-f006:**
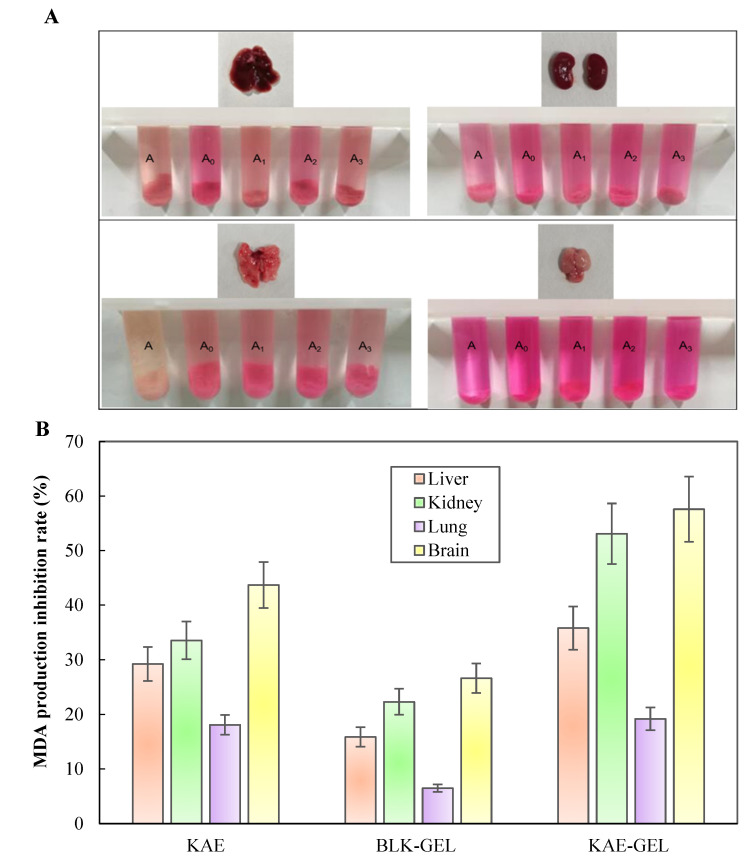
KAE and KAE-GEL inhibited the production of MDA from the tissues of mouse ex vivo. (**A**) Color shows the production of MDA in various tissues. A represents negative control group, A_0_ represents positive control group, and A_1-3_ represents KAE, BLK-GEL, and KAE-GEL. (**B**) Inhibitory rate of 0.1 mg/mL KAE and KAE-GEL on MDA production in tissues.

**Figure 7 polymers-14-03205-f007:**
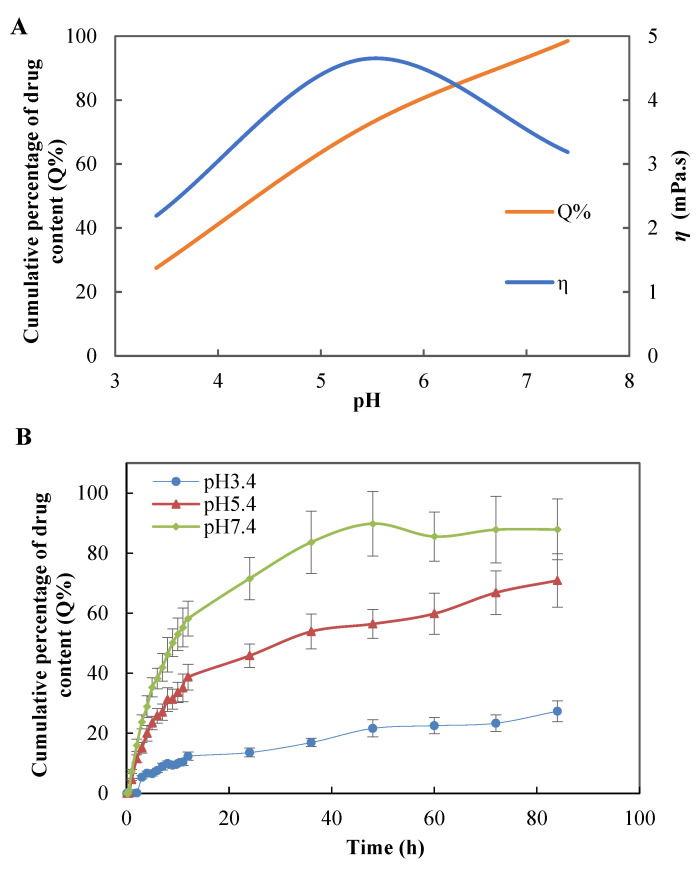
(**A**) Viscosity of hydrogel and final cumulative permeation rate at different pH values. (**B**) In vitro diffusion study of KAE-GEL through dialysis membrane (MW: 8000–14,000) at different pH values at 37 °C.

**Table 1 polymers-14-03205-t001:** Characteristics of KAE-GEL.

Samples	BLK-GEL	KAE-GEL
	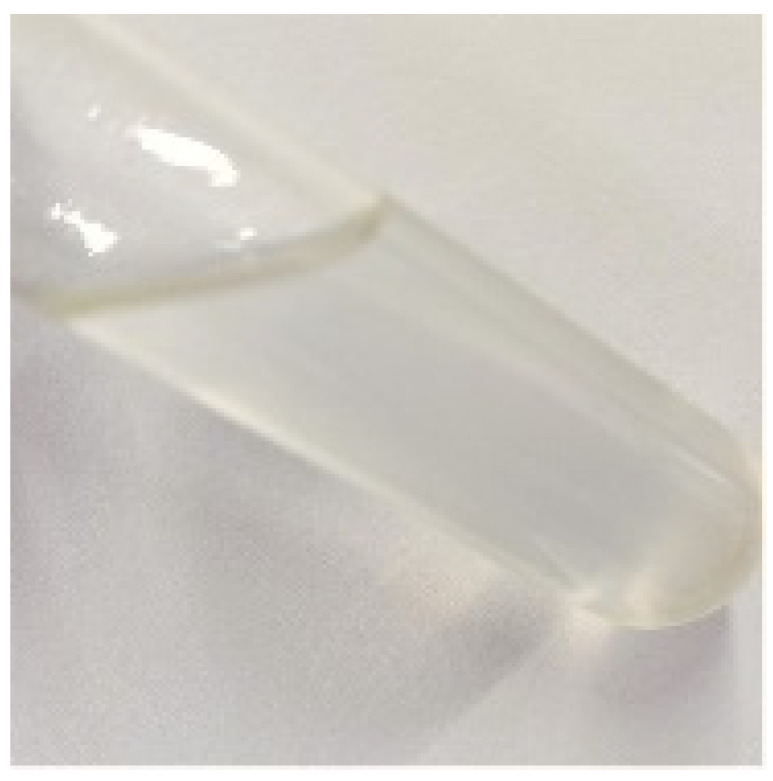	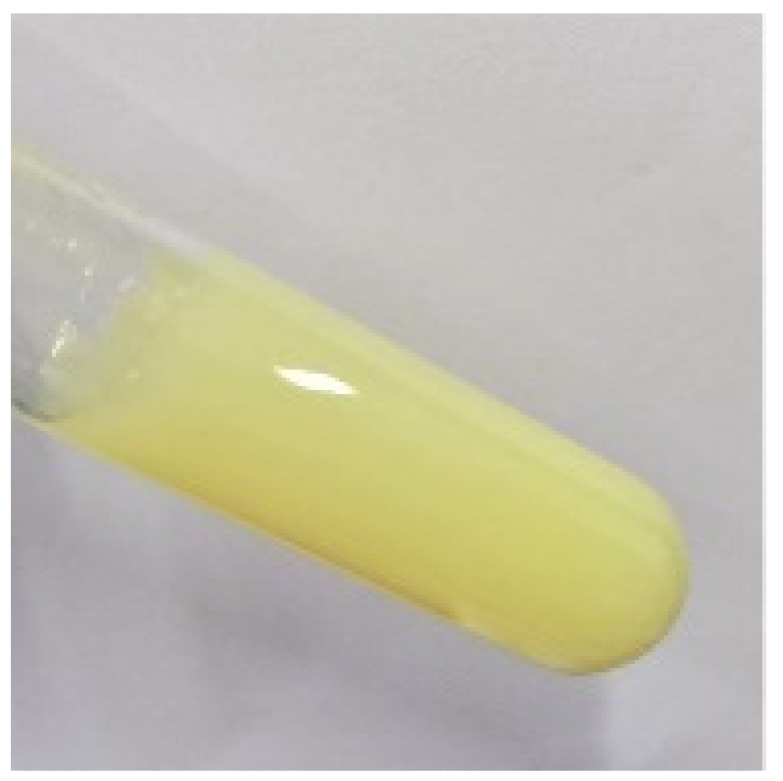
Color	colorless	bright yellow
Viscosity (mPa·s)	2.6661	3.2384

**Table 2 polymers-14-03205-t002:** Kinetic study of the in vitro release data of KAE-GEL.

Kinetic Models	pH 3.4	pH 5.4	pH 7.4
	R^2^	R^2^	R^2^
Zero-Order	0.8598	0.7903	0.6879
First-Order	0.9579	0.9770	0.9955
Higuchi	0.9694	0.9492	0.8908
Hixson-Crowell	0.7432	0.6849	0.8986

## Data Availability

Data is contained within the article.

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
