# Peer review of "In Vitro Evaluation of Kaempferol-Loaded Hydrogel as pH-Sensitive Drug Delivery Systems"

_polymers, 2022, doi:10.3390/polym14153205_

Round 1
Reviewer 1 Report
This article is about the release model of the kaempferol-loaded hydrogel. The effects of structure and compositions were studied.
1. there should not be figure 1 - this may be more proper in the results and discussion sections
2. the introduction may also briefly review the kinetics models in the current state of the art, including the characterization methods in literature reports
3. would the author expand the ph values for the release kinetics? why or why not?
Author Response
Response to Reviewer 1 Comments
We sincerely thank you for giving us an opportunity to revise the manuscript. We appreciate your valuable feedback that we have used to improve the quality of our manuscript entitled “Investigating on the smart release model with pH and antioxidant effect of kaempferol-loaded hydrogel in vitro based on its structure and composition”. We have studied comments carefully and have made correction which we hope meet with approval. Revised portions are marked with red in the paper. The main corrections in the paper and the responses to your comments are as follows:
This article is about the release model of the kaempferol-loaded hydrogel. The effects of structure and compositions were studied.
- There should not be figure 1 - this may be more proper in the results and discussion sections
Response 1: Thanks for your constructive suggestion. We have put this figure in the results and discussion section, and this figure becomes Figure 3 in the paper.
- The introduction may also briefly review the kinetics models in the current state of the art, including the characterization methods in literature reports
Response 2: Thanks for your suggestion. After reviewing the related research articles in the current state of the art, their kinetic models were described in materials and methods, so we reviewed some models in methods (lines 201-223).
- Would the author expand the pH values for the release kinetics? why or why not?
Response 3:Thanks for your suggestion. Since disease-related pH values were generally neutral or acidic, the kinetics of drug release at basic pH values were not studied. The purpose of this study was to investigate the drug release behavior of kaempferol-loaded hydrogel in acidic and neutral environments, so as to provide some basic research for future clinical applications.
Thank you very much.
Reviewer 2 Report
The document investigating on the Smart release model with pH and antioxidant effect of kaempferol-loaded hydrogel in vitro based on its structure and composition, is an interesting manuscript that fits on the topics presented in the polymers journal. Although is well presented, there are some comments that need to be addressed prior publication.
The title indicates that the release model presented is based on the structure and composition of the hydrogel, nevertheless there is few information about structure, some physicochemical and structural analysis are needed.
In the results and discussion section a table with characteristics of the KAE-gel is presented, but it is necessary to indicate which type of characteristics are presented, I suppose that are physicochemical, but they are also needed to be indicated in the materials and methods section. Additionally, there are several techniques to color determinations.
Figure 4 and 5 can be combined as there are talking about radical scavenging determinations, and in figure 5 the color change of KAE-gel it is not necessary to present it.
In general, the document is interesting, but the novelty has to be highlighted in abstract, introduction and conclusion sections.
Author Response
Response to Reviewer 2 Comments
On behalf of my co-authors, we sincerely thank you for giving us an opportunity to revise the manuscript. We appreciate your valuable feedback that we have used to improve the quality of our manuscript entitled “Investigating on the smart release model with pH and antioxidant effect of kaempferol-loaded hydrogel in vitro based on its structure and composition”. We have studied comments carefully and have made correction which we hope meet with approval. Revised portions are marked with red in the paper. The main corrections in the paper and the responses to your comments are as follows:
The document investigating on the Smart release model with pH and antioxidant effect of kaempferol-loaded hydrogel in vitro based on its structure and composition, is an interesting manuscript that fits on the topics presented in the polymers journal. Although is well presented, there are some comments that need to be addressed prior publication.
- The title indicates that the release model presented is based on the structure and composition of the hydrogel, nevertheless there is few information about structure, some physicochemical and structural analysis are needed.
Response 1: Thanks for your good suggestion. We added the intermolecular interaction process of kaempferol-loaded hydrogel formation, and the mechanism of formation between them was that kaempferol provides the hydrogen ion and carbopol hydrogel provided the carboxyl group, and the two combine to form a hydrogen bond ( Figure 3B).
- In the results and discussion section a table with characteristics of the KAE-gel is presented, but it is necessary to indicate which type of characteristics are presented, I suppose that are physicochemical, but they are also needed to be indicated in the materials and methods section.
Response 2: Thanks for your constructive suggestion. The viscosity of a hydrogel is a physicochemical property and the measurement of kaempferol-loaded hydrogel viscosity was added to the materials and methods in this paper (lines 126-135).
- Additionally, there are several techniques to color determinations. Figure 4 and 5 can be combined as there are talking about radical scavenging determinations, and in figure 5 the color change of KAE-gel it is not necessary to present it.
Response 3:Thanks for your suggestion. I have combined Figure 4 and Figure 5 into Figure 4, and also deleted the color change diagram of kaempferol-loaded hydrogel.
- In general, the document is interesting, but the novelty has to be highlighted in abstract, introduction and conclusion sections.
Response 4: Thanks for your constructive suggestion. We have highlighted the novelty of the article in the abstract, introduction and conclusion sections.
Thanks!
Reviewer 3 Report
The article deals with the preparation and characterization of kaempferol-loaded hydrogel. Although it is very interesting, there are some aspects that need to be improved:
- The abstract should be improved. The authors should clarify the aim of this study and add more results and a conclusion.
- What is the novelty of this formulation?
- The authors must clarify what is the unmet clinical need that they want to solve with this formulation. What is the target tissue? Will it be an injectable hydrogel?
- The discussion of the results is quite poor. The authors should discuss the results and compare them with other studies.
To end, some minor corrections and suggestions are attached to the PDF file.

Author Response
Response to Reviewer 3 Comments
We sincerely thank you for giving us an opportunity to revise the manuscript. We appreciate your valuable feedback that we have used to improve the quality of our manuscript entitled “Investigating on the smart release model with pH and antioxidant effect of kaempferol-loaded hydrogel in vitro based on its structure and composition”. We have studied comments carefully and have made correction which we hope meet with approval. Revised portions are marked with red in the paper. The main corrections in the paper and the responses to your comments are as follows:
The article deals with the preparation and characterization of kaempferol-loaded hydrogel. Although it is very interesting, there are some aspects that need to be improved:
- The abstract should be improved. The authors should clarify the aim of this study and add more results and a conclusion.
Response 1: We sincerely appreciate your constructive suggestion. We have revised the abstract carefully, and elaborated our work more complete.
Abstract: The purpose of this study was to prepare and evaluate kaempferol-loaded carbopol polymer (acrylic acid) hydrogel, investigate its antioxidant activity in vitro and compare the effects on drug release under different pH conditions. The kaempferol-loaded hydrogel was prepared by using carbopol 934 as the hydrogel matrix. The morphology and viscosity of the preparation were tested to understand the fluidity of the hydrogel. The antioxidant activity of the preparation was studied by scavenging hydrogen peroxide and 2,2-diphenyl-1-picrilhidrazil (DPPH) radicals in vitro and inhibiting the production of malondialdehyde in mouse tissues. The results showed that kaempferol and its preparations had high antioxidant activity. Drug release studies were conducted in three different pH media (pH 3.4, 5.4, and 7.4). In vitro release results showed that the drug release at pH 3.4, 5.4, and 7.4 was 27.32 ± 3.49ï¼…, 70.89 ± 8.91ï¼…, and 87.9 ± 10.13ï¼…. The results show that at a higher pH value (pH 7.4), greater swelling and drug release were observed.
- What is the novelty of this formulation?
Response 2: The novelty of this study was that the carbopol hydrogel matrix was used as the carrier of kaempferol to smart release kaempferol in response to different pH conditions. Carbopol hydrogel was pH-sensitive and can exhibit different swelling under different pH conditions. Kaempferol had many pharmacological effects, but it had poor hydrophobicity and low bioavailability. Therefore, this preparation was developed in this paper to improve the bioavailability of kaempferol.
Thanks for your suggestion.
- The authors must clarify what is the unmet clinical need that they want to solve with this formulation. What is the target tissue? Will it be an injectable hydrogel?
Response 3: This paper mainly discussed the antioxidant activity of kaempferol-loaded hydrogel and its release behavior under different pH conditions. Currently, no research has been conducted on the pharmacological and pharmacodynamic effects of this preparation, which our research group plans to carry out at a later stage.
Target tissue: we think that this preparation can treat some inflammation and tumor, and our laboratory will do some skin inflammation diseases later. It can be used as an injectable hydrogel, but not for vascular injection.
Thanks for your constructive suggestion.
- The discussion of the results is quite poor. The authors should discuss the results and compare them with other studies.
Response 4: The experimental results are discussed in detail and some references are given.
Thank for your suggestion.
- To end, some minor corrections and suggestions are attached to the PDF file.
Response 4: Thanks for your constructive suggestions. We have revised the paper according to the suggestions in the pdf, and responded to the questions raised in the PDF.
Thanks a lot!

Round 2
Reviewer 2 Report
The document has been clearly improved, the authors took into consideration the suggestions.
Just a phrase or a line needs to be added in the abstract to highlight the principal novelty or to indicate possible applications.
Author Response
Response to Reviewer 2 Comments
Dear Reviewer:
We are very grateful to your valuable suggestion. We have studied this comment carefully and revised it. Revised portions are marked with red in the paper. The main corrections in the paper and responses to your comments are as follows:
The document has been clearly improved, the authors took into consideration the suggestions. Just a phrase or a line needs to be added in the abstract to highlight the principal novelty or to indicate possible applications.
Response : Thanks for the reviewer’s constructive suggestion. We added a sentence in the abstract to highlight the novelty of the paper:Kaempferol-loaded carbopol hydrogel displayed greater swelling and drug release at higher pH values (pH 7.4).
Thanks a lot.
Reviewer 3 Report
Dear Authors,
Thank you for fixing the article. Although the article has been widely improved, you can find attached my last minor corrections and comments.

Author Response
Response to Reviewer 3 Comments
Dear Reviewer:
We sincerely thank you for your valuable suggestions. We have carefully studied these comments and revised them, hoping to get approval. Revised portions are marked with red in the paper. The main corrections in the paper and responses to your comments are as follows:
Dear Authors,
Thank you for fixing the article. Although the article has been widely improved, you can find attached my last minor corrections and comments.
Response : We sincerely appreciate the reviewer’s constructive suggestion. We have revised and responded according to your valuable comments in the PDF.
Thanks!
